# Sample-Efficient Co-Optimization of Agent Morphology and Policy with Self-Imitation Learning

## Abstract

The task of co-optimizing the body and behaviour of agents has been a long-standing problem in the fields of evolutionary robotics and embodied AI. Previous work has largely focused on the development of learning methods exploiting massive parallelization of agent evaluations with large population sizes, a paradigm which is applicable to simulated agents but cannot be transferred to the real world due to the assoicated costs with the production of embodiments and robots. Furthermore, recent data-efficient approaches utilizing reinforcement learning can suffer from distributional shifts in transition dynamics as well as in state and action spaces when experiencing new body morphologies. In this work, we propose a new co-adaptation method combining reinforcement learning and State-Aligned Self-Imitation Learning to co-optimize embodiment and behavioural policies withing a handful of design iterations. We show that the integration of a self-imitation signal improves the data-efficiency of the co-adaptation process as well as the behavioural recovery when adapting morphological parameters.

## 1 Introduction

Finding an optimal combination of body and morphology of agents has been a long-standing research problem, finding its roots in the community of evolutionary robotics (Lipson & Pollack, 2000; Clune et al., 2013; Doncieux et al., 2015). Originally, research in this area largely focused on the use and development of evolutionary or genetic algorithms adapting body and control parameters at the same time (Lipson & Pollack, 2000; Watson et al., 2002; Bongard, 2011; Buason et al., 2005; Kempen & Eiben, 2022). This was and is largely inspired by observations made about the evolutionary principles governing the adaptation of animal species in nature bringing forth animals with unique morphological features and behaviours, such as *Carparachne aureoflava*, a spider capable of "wheeling" down sand dunes to escape predators (Harvey & Zukoff, 2011; Western et al., 2023). More recent research (Hale et al., 2019; Luck et al., 2019) has presented evidence of the benefits of considering the different time-scales on which co-adaptation of body and behaviour occurs in the real world: adaptation of the body is costly and time-consuming, as it involves growing appendices, organs and tissue in nature; likewise in robotics, where even fast manufacturing methods like 3D-printing and casting require a considerable amount of work-hours and material. However, adaptation of behaviour occurs at much faster time-scales, enabled by fast and inexpensive changes to neurons in the brain or changes to control parameters and artificial neural network weights in robots.

Recent years have brought forward several works considering the use of reinforcement learning (RL) methods for the problem of co-adapting robots (Chen et al., 2021; Pigozzi et al., 2023; Sun et al., 2023; Luck et al., 2019), usually with a fast behavioural adaptation process and slower morphology adaptation. This allowed to develop methods capable of being deployed in principle on real-world robotics due to their data-efficiency. However, data-efficient co-adaptation processes can suffer considerably from the problem of distributional shift inherent to the co-adaptation problem setting. Every new agent morphology the algorithms experiences brings with it changes to the transition distribution, as well as to the semantics of state and action spaces. For example, changes to the orientation of a robot leg lead to changes between the mapping of motor actions and of orientation and movement of the robot leg. This can be detrimental to the co-adaptation process, as changes to

the morphology can lead to catastrophic forgetting due to policy actions causing different motion patterns between individual designs.

We propose a novel co-adaptation methodology tackling the aforementioned problems by combining reward-driven reinforcement learning and self-imitation learning utilizing Wasserstein distances for data-efficient adaptation of body and behaviour of agents. The idea of our approach is to not only force the reinforcement learning algorithm to adapt body and behaviour for maximizing an objective function such as forward velocity, but also to encourage the imitation of the agent's 'ancestors' and their previous behaviours to increase learning stability and accelerate the co-adaptation progress.

In this paper[1], we present the following contributions:
**(C1)** An extension of State-Alignment Imitation Learning (SAIL) (Liu et al., 2019) for mismatching morphologies to State-Aligned Self-Imitation Learning for the problem of co-adapting the morphology and behaviour of agents.
**(C2)** A novel co-adaptation method, **Co**-Adaptation with **S**elf-**I**mitation **L**earning (CoSIL), utilizing State-Aligned Self-Imitation Learning to optimize an agent's morphology and behaviour data-efficiently on fewer design iterations.
**(C3)** We demonstrate in an empirical study the benefits and limitations of CoSIL by evaluating its performance versus a non-self-imitating baseline in a range of locomotion tasks.

## 2 BACKGROUND

**Reinforcement Learning (RL):** In a reinforcement learning setting, problems are formulated as a Markov decision process (MDP) $\langle S, A, r, p \rangle$. We consider an environment-agent interaction fully described by a set of possible states $S \in \mathbb{R}^m$, a set of possible actions taken by the agent in a given state $A \in \mathbb{R}^n$, a reward function $r : S \times A \mapsto \mathbb{R}$ and a transition function $p : S \times A \times R \times S \mapsto [0, 1]$. The transition function defines the dynamics of the environment by providing a probability $p(s'|s, a)$ of each next state given the current state and the chosen action. In order to train an agent for a given task, we model the desired behaviour as a reward function and use an optimization procedure to design a policy $\pi(a|s) \in [0, 1]$ which approximates the optimal action $a$ to take in any given state $s$ as a probability distribution over $A$ to maximize the cumulative rewards.

**Multi-Body Reinforcement Learning:** In multi-body reinforcement learning, we consider an extension to the classic Markov Decision Process (MDP) suitable for modelling the fact that both behaviour and morphological parameters are adapted. The Multi-Body MDP (MB-MDP) consists of $(S, A, \Xi, r, p(s_{t+1}|s_t, a_t, \xi), p(s_0|\xi))$ with state space $S \in \mathbb{R}^s$ and action space $A \in \mathbb{R}^a$. Notably, in a MB-MDP the set $\Xi$ models the morphological parameter space, containing individual instances of agent morphologies $\xi \in \Xi$. Throughout this paper, we will without a loss of generality consider $\Xi \in \mathbb{R}^d$ for $d$ continuous design parameters, such as limp lengths or width/size of agent body elements. As changes to the physics of the agent morphology impact its dynamics, the transition function $p(s_{t+1}|s_t, a_t, \xi)$ depends on the current morphology parameter $\xi$. The reward function $r(s_t, a_t, \xi)$ may also implicitly depend on $\xi$ via the transition function, or explicitly if the manufacturing costs are taken into account, for example. The objective is to find a policy $\pi_\theta(s_t, \xi) = a_t$ which maximizes the finite-horizon expected discounted reward

$$R(\xi, \pi) = \mathbb{E}_{\substack{s_{t+1} \sim p(s_{t+1}|s_t, a_t, \xi) \\ s_0 \sim p(s_0|\xi) \\ a_t \sim \pi(s_t, \xi)}} \left[ \sum_{t=0}^{T} \gamma^t r(s_t, a_t, \xi) \right] \tag{1}$$

given an embodiment $\xi$, the policy $\pi$, and discount factor $\gamma \in (0, 1)$.

**Co-Adaptation of Agent Body and Behaviour:** The previous formalism allows us to formulate the joint optimization of behaviour and morphology of agents as

$$\pi^*, \xi^* = \arg \max_\xi \max_\pi R(\xi, \pi); \tag{2}$$

in other words, we are interested in finding both the optimal morphology $\xi^*$ and optimal policy $\pi^*$ given a reward function $r(s_t, a_t, \xi)$. If we consider the semantics of the parameters and the

---

[1]Supplemental material can be found at `url-removed-for-anonymity`

optimization time-scales (i.e., policy learning can be done faster than morphology adaptation), this problem can be considered a bi-level optimization problem. Given the current morphology of the agent in the inner optimization problem, we can solve the RL problem using Eq. (1). In the outer optimization problem, given performances $R(\xi, \pi)$ of past morphology-policy pairs $(\xi_i, \pi_i)$, we can again utilize optimization methods or reinforcement learning to find new candidate morphologies $\xi$ to evaluate.

## 3 CO-ADAPTATION WITH SELF-IMITATION LEARNING

In this section, we will first introduce the individual components of ***Co-Adaptation with Self-Imitation Learning (CoSIL)*** using State-Aligned Imitation Learning (SAIL) (Liu et al., 2019). We will end the section with a description of the main algorithm.

### 3.1 SELF-IMITATION LEARNING ON CO-ADAPTATION SEQUENCES

Assume a MB-MDP $(S, A, \Xi, r, p(s_{t+1}|s_t, a_t, \xi), p(s_0|\xi))$, as given in Section 2. Naturally, a co-adaptation process will produce a sequence of morphology-policy tuples $\{(\xi_0, \pi_0), (\xi_1, \pi_2), (\xi_3, \pi_3), \cdots\}$. Given two morphology-policy pairs $(\xi_i, \pi_i)$ and $(\xi_j, \pi_j)$, we can formulate the trajectory distributions

$$q(\tau^i) = p(s_0|\xi_i) \prod_{t=0}^{T-1} p(s_{t+1}|s_t, a_t, \xi_i)\pi_i(a_t|s_t, \xi_i) \tag{3}$$

and

$$p(\tau^j|\pi_j) = p(s_0|\xi_j) \prod_{t=0}^{T-1} p(s_{t+1}|s_t, a_t, \xi_j)\pi_j(a_t|s_t, \xi_j). \tag{4}$$

We will now assume that the pair $(\xi_i, \pi_i)$ represents our expert, that is, the training on morphology $\xi_i$ has concluded and $\pi_i$ has learned an optimal movement strategy for $\xi_i$ (i.e., $\pi_i^*|\xi_i$). If we are now currently training on morphology $\xi_j$, where $j > i$, then we can force the policy $\pi_j$ to imitate the previous agent by optimizing

$$\min_{\pi_j} \mathcal{D}(q(\tau^i), p(\tau^j|\pi_j)), \tag{5}$$

for a divergence measure $\mathcal{D}$ expressing the distance between these two probability distributions. Importantly, we consider here that $\xi_j$ is fixed and not optimized, otherwise $(\xi_i, \pi_i)$ is a trivial solution to this problem. While different choices exist for this divergence measure, we will follow state alignment-based imitation learning and use state-distribution matching via generative adversarial learning.

### 3.2 FEATURE-STATE-DISTRIBUTION SELF-IMITATION LEARNING

As previously described, a core problem for imitation learning between agents with different body morphologies is that the semantic of state and action spaces can shift considerably. If in one agent morphology the motor action of 1.0 may lead to moving a limp upwards, in another morphology it may cause it to go to the side, even if both agents are in the exact same state. Hence, using the original state and action spaces are not necessarily suitable to use in imitation learning. Therefore, we assume in the following a function $\phi: S \to S^{F}$ [2] which maps the state of the agent to a shared feature space $S^F$. In practice, such a feature space could be image-based or, as used in this paper, based on motion capture markers placed on the body.

In our proposed self-imitation learning approach for co-adaptation, we are matching the state distributions between previous expert behaviour and the current agent, a technique used successfully in prior work (Fickinger et al., 2021; Rajani et al., 2023). Similarly, we use the marginal feature-space state distributions for the expert trajectories from past morphologies

$$q(\phi(s)) = \mathbb{E}_{\substack{s_{t+1}\sim p(s_{t+1}|s_t, a_t, \xi_i) \\ a_t \sim \pi_i(a_t|s_t, \xi_i) \\ s_0 \sim p(s_0|\xi_i)}} \left[ \frac{1}{T} \sum_{t=0}^{T} \mathbb{1}(\phi(s_t) = \phi(s)) \right] \tag{6}$$

---

[2]Note, that we use without loss of generality $\phi: S \to S^F$ for better readability and clarity. However, $\phi: S \times \Xi \to S^F$ would be more accurate as the mapping also depends on the current embodiment of the agent.

and for the current agent morphology

$$p(\phi(s)|\pi_j) = \mathbb{E}_{\substack{s_{t+1} \sim p(s_{t+1}|s_t, a_t, \xi_j) \\ a_t \sim \pi_j(a_t|s_t, \xi_j) \\ s_0 \sim p(s_0|\xi_j)}} \left[ \frac{1}{T} \sum_{t=0}^{T} \mathbb{1}(\phi(s_t) = \phi(s)) \right], \tag{7}$$

with $\mathbb{1}$ being a Kronecker delta function, returning the value 1 iff $\phi(s_t) = \phi(s)^3$ holds true and 0 otherwise. Using these state distributions we can now reformulate Eq. (5) with

$$\mathcal{D}(q(\phi(s)), p(\phi(s)|\pi_j)), \tag{8}$$

where we can use divergences such as Kullback-Leibler's, the Wasserstein distance, or the Jensen-Shannon divergence. Eq. (8) will be our main objective for enabling self-imitation learning across morphologies.

### 3.3 IMITATION REWARD AND ENVIRONMENTAL REWARD

CoSIL makes use of two reward functions: $r^{\text{IL}}$ for the self-imitation reward, and $r^{\text{RL}}$ for the environment reward we aim to maximize as the main objective. While $r^{\text{RL}}$ is a fixed objective given by the environment, $r^{\text{IL}}$ is a learned function which rewards the agent for a behavioural policy $\pi$ minimizing Eq. (8), given a demonstration dataset $\tau^{\text{E}}$. Multiple choices exist for the imitation learning method used to learn $r^{\text{IL}}$. Candidates include the Adversarial Inverse Reinforcement Learning (AIRL) reward

$$r^{\text{IL}}(\phi(s_t), \phi(s_{t+1})) = \log(\rho(\phi(s_t))) - \log(1 - \rho(\phi(s_t))), \tag{9}$$

where $\rho$ is a discriminator which differentiates between agent states and expert states, as well as State-Aligned Imitation Learning (SAIL) using the Wasserstein distance with reward function

$$r^{\text{IL}}(\phi(s_t), \phi(s_{t+1})) = \rho(\phi(s_{t+1})) - \mathbb{E}_{s \sim \tau^{\text{E}}} \left[ \rho(\phi(s)) \right], \tag{10}$$

where $\rho$ is a learned discriminator function (i.e., a neural network) modelling the Kantorovich's potential, assigning higher values to states similar to those seen in the expert dataset $\tau^{\text{E}}$. Further details about the training procedure to learn these reward functions can be found in (Fu et al., 2018) for AIRL, as well as (Liu et al., 2019) for SAIL. In this paper, we will consider mainly the SAIL reward in Eq. (10), as previous work has shown it performs better in this task setting (Rajani et al., 2023).

### 3.4 POLICY LEARNING WITH SELF-IMITATION LEARNING

CoSIL makes use of Soft Actor Critic (SAC) (Haarnoja et al., 2018) as the reinforcement learning backbone of the method with a slight adaptation to the learning rule for policy updates. As we have two reward functions, $r^{\text{RL}}$ as the original objective and $r^{\text{IL}}$ as the self-imitation reward, we propose to adapt SAC to learn two Q-functions with

$$\mathcal{L}_{Q_k^{\text{RL}}} = \frac{1}{2}(Q_k^{\text{RL}}(s_t, a_t, \xi) - (r^{\text{RL}}(\phi(s_t), \phi(s_{t+1})) + \gamma(\min_{k=1,2} Q_k^{\text{RL}}(s_{t+1}, a_{t+1}, \xi) - \alpha \log(\pi(a_{t+1}|s_{t+1}, \xi)))))^2, \tag{11}$$

$$\mathcal{L}_{Q^{\text{IL}}_k} = \frac{1}{2}(Q_k^{\text{IL}}(s_t, a_t, \xi) - (r^{\text{IL}}(\phi(s_t), \phi(s_{t+1})) + \gamma(\min_{k=1,2} Q_k^{\text{IL}}(s_{t+1}, a_{t+1}, \xi) - \alpha \log(\pi(a_{t+1}|s_{t+1}, \xi)))))^2. \tag{12}$$

Since both reward functions can differ in magnitude and to avoid imbalances during training, we normalize both rewards using z-score normalization. This leads to the following loss function for the policy $\pi$ with two Q-networks:

$$\mathcal{L}_\pi = (1 - \omega) \min_{k=1,2} Q_k^{\text{RL}}(s_t, a_t, \xi) + \omega \min_{k=1,2} Q_k^{\text{IL}}(s_t, a_t, \xi) - \alpha \log \pi(a_t \mid s_t, \xi), \tag{13}$$

in which we optimize the policy both for the objective-driven Q-function $Q_{\text{RL}}$ and the self-imitation Q-function $Q_{\text{IL}}$, weighted by the parameter $\omega$. Each of the critics uses the double-Q trick proposed by (Hasselt, 2010), by which the minimum output of an ensemble of two neural networks is taken as the critic's output.

---

$^3$Note, that of course in continuous state spaces we measure if $\phi(s)$ is in a sphere of diameter $\epsilon$ around $\phi(s_t)$.

### 3.5 MORPHOLOGY OPTIMIZATION

Similar to the behaviour learning process, we extend the morphology optimization objective to incorporate self-imitation. Accordingly, we supplement the objective introduced in (Luck et al., 2019) by adding the Q-function $Q_j^{\text{IL}}$ with

$$\max_{\xi} \mathbb{E}_{s_0 \sim p(s_0|\xi)} [(1 - \omega_{\text{opt}}) \min_{j=1,2} Q_j^{\text{RL}}(s_0, \pi_{pop}(a_0|s_0, \xi), \xi) + \omega_{\text{opt}} \min_{j=1,2} Q_j^{\text{IL}}(s_0, \pi_{pop}(a_0|s_0, \xi), \xi)], \tag{14}$$

where $\omega_{\text{opt}}$ is used to weigh the importance of the self-imitation reward versus the environment reward function. While in principle any optimization method can be used, we found the gradient-free Particle Swarm Optimization (PSO) optimizer (Kennedy & Eberhart, 1995) to be the most efficient.

It is worth to note that evaluating $Q_j^{\text{RL}}$ and $Q_j^{\text{IL}}$ is computational- and data-efficient because the Q-function acts as a surrogate function, predicting the performance of a design $\xi$ based on past experience and without requiring simulation. Since the distribution $p(s_0|\xi)$ is generally unknown, we replace it in practice with $s_0 \sim R_0$, where $R_0$ is a replay buffer containing only starting states. This approach also increases the real-world applicability of the methodology.

### 3.6 CO-DESIGN WITH SELF-IMITATION LEARNING

We present the proposed CoSIL method in Algorithm 1. Two replay buffers are employed in our system: a buffer $\mathbf{C}$ containing only observations collected from the current morphology, and a buffer $\mathbf{P}$ containing observations obtained from previous designs. As proposed in (**?**), we then use two instances of the previously introduced SAC algorithm, each with its own set of actor and critic networks: a population agent which is trained offline after each morphology change with observations from $\mathbf{P}$ and an individual agent which is trained online using observations from $\mathbf{C}$. Every time a new morphology is selected for evaluation, the individual agent is initialized by copying the network parameters from the population agent. We refer to the poli-

---

**Algorithm 1 Co-Adaptation with Self-Imitation Learning (CoSIL)**

**Input:** $\mathbf{D}^{\text{E}} = [\tau_0^{\text{E}}, ...], r^{\text{RL}}$ and $p$
1: Initialize $\pi_{\text{ind}}, \pi_{\text{pop}}, Q_{\text{ind}}^{\text{RL}}, Q_{\text{ind}}^{\text{IL}}, Q_{\text{pop}}^{\text{RL}}, Q_{\text{pop}}^{\text{IL}}$ and $r^{\text{IL}}$
2: $\xi \leftarrow \xi_0, \Xi \leftarrow \varnothing, \mathbf{P} \leftarrow \varnothing, \mathbf{C} \leftarrow \varnothing, \mathbf{D} \leftarrow \mathbf{D}^{\text{E}}$
3: **while** not converged **do**
4:     **for** $e = 1, ..., E$ **do**
5:         Sample $\mathbf{s}_0$ from the environment
6:         Sample a trajectory
        $\tau_{e,\xi} = (\mathbf{s}_0, \pi_{\text{ind}}(a_0|s_0, \xi), \mathbf{s}_1, \cdots)$
7:         Add $\{\mathbf{s}_t, \mathbf{a}_t, r^{\text{RL}}(\mathbf{s}_t, \mathbf{a}_t, \xi), \mathbf{s}_{t+1}, \xi\}$ to $\mathbf{C}$
8:         Sample a batch $B$ from $\mathbf{C}$
9:         Update $r^{\text{IL}}$, given $B$ and $\mathbf{D}$
10:       Update $Q_{\text{ind}}^{\text{RL}}$ and $Q_{\text{ind}}^{\text{IL}}$, given $B$ and $r^{\text{IL}}$
11:       Update $\pi_{\text{ind}}$ as in Eq. (13), given $B$ and $\omega_{\text{ind}}$
12:     **end for**
13:     Add the observation $o$ to $\mathbf{P}, \forall o \in \mathbf{C}$
14:     **for** $u = 1, ..., U_{\text{pop}}$ **do**
15:         Sample a batch $B$ from $\mathbf{P}$
16:       Update $Q_{\text{pop}}^{\text{RL}}$ and $Q_{\text{pop}}^{\text{IL}}$, given $B$ and $r^{\text{IL}}$
17:       Update $\pi_{\text{pop}}$ as in Eq. (13), given $B$ and $\omega_{\text{pop}}$
18:     **end for**
19:     $\pi_{\text{ind}} \leftarrow \pi_{\text{pop}}, Q_{\text{ind}}^{\text{RL}} \leftarrow Q_{\text{pop}}^{\text{RL}}$ and $Q_{\text{ind}}^{\text{IL}} \leftarrow Q_{\text{pop}}^{\text{IL}}$
20:     Add $\{\xi, [\tau_{1,\xi}, ..., \tau_{E,\xi}]\}$ to $\Xi$
21:     $\xi \leftarrow$ Morph-Opt$(\mathbf{P}, \Xi, Q_{\text{ind}}^{\text{RL}}, Q_{\text{ind}}^{\text{IL}})$ with Eq. (14).
22:     Re-select the demonstrations $\mathbf{D}$
23:     $\mathbf{C} \leftarrow \varnothing$
24: **end while**

---

cies and critics belonging to the population and individual agents with the subscripts $pop$ and $ind$, respectively. This approach has been described by (Luck et al., 2019) to increase data-efficiency and performance of reinforcement-learning-driven Co-Adaptation. The number of episodes used to train online under each design is denoted as $E$, while $U_{\text{pop}}$ refers to the fixed amount of offline updates to the population agent. $\mathbf{D}^{\text{E}}$ refers to the initial expert observations, and $\mathbf{D}$ denotes the set of demonstrations selected from previous morphologies for their optimal behavior using a selection-heuristic. The heuristic we use to update the demonstration dataset in line 22 is to replace the 30% of worst performing trajectories in $\mathbf{D}$ with an equal number of best performing trajectories from the last ten episodes, if the latter's episodic return is higher. Morph-Opt refers to the design optimization procedure using PSO with the objective function presented in Eq. (14).

## 4 EXPERIMENTS

To understand the potential benefits and impact of using a self-imitation learning signal in the co-adaptation setting we empirically evaluate CoSIL in a number of continuous control experiments with adaptable design parameters. Due to the time, cost and resource constraints we focus primarily

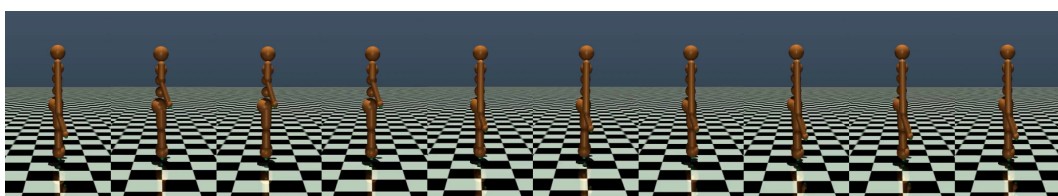

Figure 1: Designs in the HalfCheetah environment evolved by CoSIL, from left to right and continuing on the second row. The sequence of designs was obtained from a randomly chosen seed.

Figure 2: Designs in the Humanoid environment evolved by CoSIL, from left to right. The sequence of designs was obtained from a randomly chosen seed.

on evaluations in simulation in this paper, however, with a particular interest in potential benefits for data-efficiency to allow for real-world robotic experiments in the future. In particular, we set out to investigate the following research questions:

**(RQ1)** Is the use of self-imitation learning advantageous when co-optimising the behaviour and morphology of agents and robots for a given environmental reward ($r^{\text{RL}}$)?

**(RQ2)** What are the limitations of the approach? Is self-imitation learning always beneficial?

**(RQ3)** How does self-imitation compare against pure imitation learning for co-adaptation?

## 4.1 EXPERIMENTAL SETUP

In our experiments, we used variants of the OpenAI Gym library (Brockman et al., 2016) environments Humanoid, Walker and HalfCheetah adapted to the co-adaptation setting, as previously proposed (Rajani et al., 2023). These environments are implemented using the MuJoCo physics engine (Todorov et al., 2012). Experiments are conducted on a computing cluster with GPU models NVIDIA RTX4500. We employed 32GB of RAM and were constrained by 72 hours of real time usage per experiment. The results are averaged across four distinct seeds. For both baselines and CoSIL we start the training process from an initial training set (i.e., replay buffer) containing the experience of five randomly sampled designs trained for the same number of episodes, for which standard SAC was used. Similarly, the initial demonstration dataset for CoSIL was generated from a trained expert

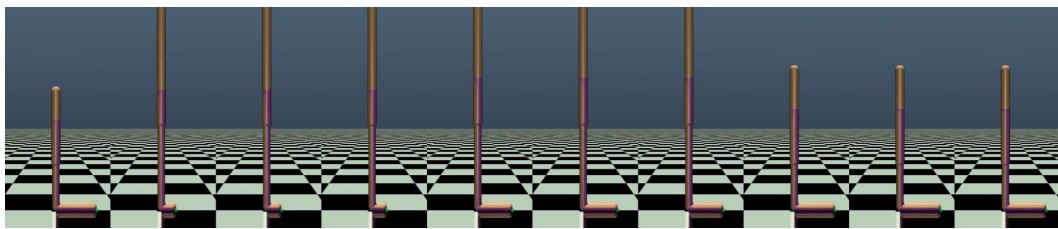

Figure 3: Designs in the Walker environment evolved by CoSIL, from left to right. The sequence of designs was obtained from a randomly chosen seed.

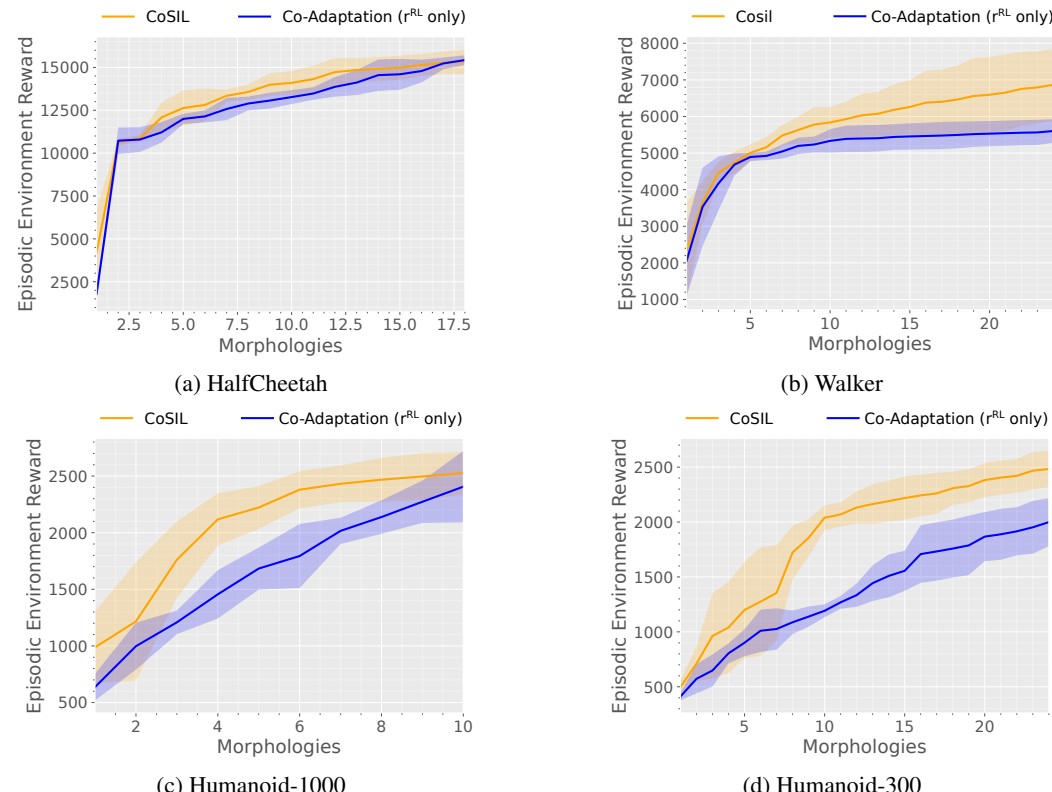

Figure 4: Comparison between our proposed approach CoSIL ($r^{\text{IL}}$ and $r^{\text{RL}}$) and Co-Adaptation (Luck et al., 2019) ($r^{\text{RL}}$ only) on the four tasks HalfCheetah, Walker, Humanoid-1000 and Humanoid-300 in MuJoCo. Plots show the performance of each morphology measured by averaging the 20% best episodes, and arranging the order of the morphologies by performance along the x-axis (see Appendix for plots without ordering). Experiments were repeated four times with distinct seeds. While each algorithm was trained for 1000 episodes on Humanoid-1000, in Humanoid-300 only 300 episodes were used. Comparing Fig. (c) and (d) shows that CoSIL increases the data-efficiency considerably when allowing for less episodes per morphology.

policy of a randomly selected design. Furthermore, a first experiment on a simulated Unitree Go1 robot can be found in Appendix D.

## 4.2 SELF-IMITATION LEARNING FOR CO-OPTIMIZATION OF AGENT DESIGN AND BEHAVIOUR

First, we evaluate the general efficiency of **Co**-Adaptation with **S**elf-**I**mitation **L**earning (CoSIL) over a standard co-adaptation algorithm (Co-Adaptation) (Luck et al., 2019) using only the environmental reward function $r^{\text{RL}}$ (RQ1). For this, we evaluate CoSIL and Co-Adaptation in three environments, namely HalfCheetah, Walker and Humanoid. As we can see in the results presented in Figure 4, the use of both self-imitation reward $r^{\text{IL}}$ and environmental reward $r^{\text{RL}}$ generally leads to the uncovering of better performing morphologies. However, as we can see in Figure 4-4a the gap between Co-Adaptation and CoSIL is relatively small in simpler tasks such as HalfCheetah, while CoSIL noticeable outperforms the baseline in tasks such as Walker and Humanoid which require a larger amount of coordination and reflexes to maintain the pose of the agent. Thus, we conclude that it is not always beneficial to combine Co-Adaptation with a self-imitation training signal, which is associated with a higher cost of computation (RQ2). Self-imitation seems to be especially beneficial in tasks of higher complexity and difficulty: noticeably, in Walker (Fig. 4-4b) CoSIL uncovers considerably better performing morphologies than Co-Adaptation, outperforming the latter by a large margin.

In Figure 1, we present sample images taken of ten morphologies evolved by CoSIL for a randomly chosen seed in the HalfCheetah environment. The evolution process can be followed from left to

Table 1: Average performance of CoSIL and three baselines on the Walker task. CoSIL (no-update) does not update the set of past expert demonstrations; Coadapt ($r^{\text{RL}}$ only) (Luck et al., 2019) uses only the environmental reward; COIL ($r^{\text{IL}}$ only) (Rajani et al., 2023) uses only the imitation reward.

|           | CoSIL       | Coadapt  | CoSIL (no update) | COIL    |
|-----------|-------------|----------|-------------------|---------|
| Design 1  | **2340.06** | 2072.92  | **2340.06**       | 105.65  |
| Design 5  | **5027.85** | 4888.67  | 4866.31           | 4323.15 |
| Design 10 | **5897.35** | 5340.30  | 5712.51           | 4837.46 |
| Design 15 | **6237.85** | 5460.68  | 5951.12           | 4971.22 |
| Design 20 | **6599.13** | 5546.25  | 6053.81           | 5112.78 |
| Design 24 | **6851.80** | 5608.66  | 6107.24           | 5151.46 |

right, where the second row of designs follows after the first. Similarly, in Figures 2 and 3, we present the same visualisations for the Humanoid and Walker environments, respectively.

### 4.3 INCREASED DATA-EFFICIENCY

Furthermore, we investigate the impact of self-imitation learning on data-efficiency in the most difficult Humanoid task (RQ1). For this we perform two experiments in which both CoSIL and Co-Adaptation optimize behaviour and morphology, in one experiment allowing for only 300 episodes per morphology (Fig. 4-4d), and in another for 1000 episodes (Fig. 4-4c). It is evident from this experiment that while CoSIL suffers from some performance degradation in the initial designs, the discovery of high performing morphologies and behaviours is largely undisturbed in the later training stage. On the other hand, Co-Adaptation suffers considerably from a shorter amount of training time on morphologies (Fig. 4-4d), and is not able to recover and discover similar performing morphologies and behaviours than with more training data (Fig. 4-4c).

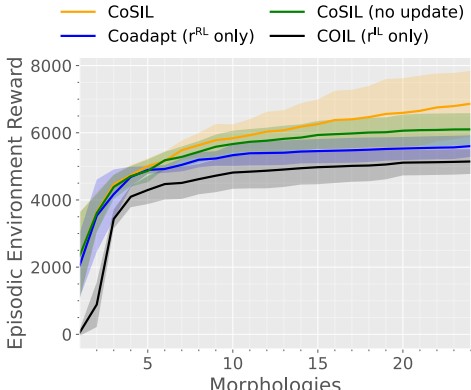

Figure 5: Comparison of the proposed method CoSIL versus baselines and ablations on the Walker task: CoSIL (no-update) does not update the set of past expert demonstrations; Coadapt ($r^{\text{RL}}$ only) (Luck et al., 2019) uses only the environmental reward; COIL ($r^{\text{IL}}$ only) (Rajani et al., 2023) uses only the imitation reward. It can be seen that the proposed method outperforms the baselines and ablation.

### 4.4 SELF-IMITATION LEARNING VERSUS IMITATION LEARNING FOR CO-ADAPTATION

In this study we investigate in particular the performance differences of using self-imitation learning versus standard imitation learning for the co-adaptation of design and behaviour. Specifically, we compare the use of self-imitation learning with two previous approaches, namely Co-Adpatation (Luck et al., 2019) and COIL (Rajani et al., 2023). As already mentioned, Co-Adaptation (Luck et al., 2019) optimizes solely for the environmental reward $r^{\text{RL}}$. COIL (Rajani et al., 2023) on the other hand uses only an imitation reward $r^{\text{IL}}$ derived from a fixed set of expert demonstrations. Furthermore, we compare to a version of CoSIL in which we do not update the set of demonstrations, i.e., we only perform imitation learning and no self-imitation learning by using only the initial set of expert demonstrations, which we name *CoSIL (no update)*. However, this version of CoSIL still uses both imitation reward $r^{\text{IL}}$ and environmental reward $r^{\text{RL}}$, which positions it methodological between CoSIL and COIL. The comparison between these approaches on the Walker task can be found in Figure 5 and in Table 1. As expected, the pure imitation learning approach from expert demonstrations COIL (black) reaches an overall lower performance, as it is not directly optimizing for the environmental reward. On the other hand, using the proposed approach without self-imitation learning by not updating the set of demonstrations leads to a better performance that standard Co-Adpatation using environmental rewards, but is outperformed by the proposed approach utilizing self-imitation learning.

### 4.5 IMPACT OF FEATURE-SELECTION

We perform an additional experiment evaluating the impact the selection of features to match with self-imitation learning has on CoSIL. For this we evaluate CoSIL on the HalfCheetah task while using two distinct sets of features for the self0imitation process. Specifically, we train CoSIL using features extracted from markers at bot the knee and foot of HalfCheetah, while the second approach uses only foot markers. In both cases, we extract the velocity and height-normalised position relative to the base joint for each marker, and use these as morphology-independent features. As can be seen in Figure 6 the selection of the feature set has a clear impact on the performance of CoSIL. Furthermore we can note that indeed a minimal set of features, here the features extracted from the foot marker, leads to a better performance. We hypothesise that this allows for a better imitation learning agnostic to the specific morphological parameters, imposing less restrictions to the possible movements the policy can learn to maximize the environmental reward.

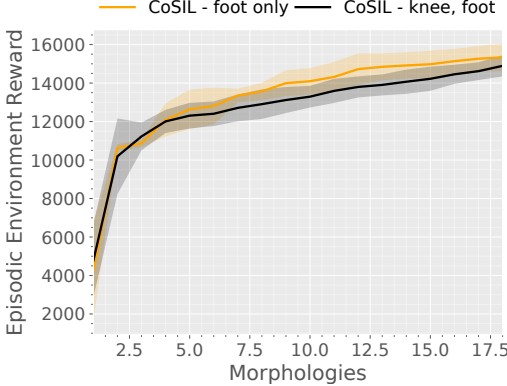

Figure 6: Evaluation of the impact of marker selection in the HalfCheetah task: *CoSIL - foot only* uses only foot markers, while *CoSIL - knee,foot* uses the knee marker in addition. It can be seen that marker selection has a clear impact on performance, and in fact using too many markers impacts the performance of CoSIL negatively.

## 5 RELATED WORK

**Evolutionary Robotics:** Designing robot hardware with evolutionary principles has been a long-standing research effort. Seminal work by (Lipson & Pollack, 2000) explored using genetic algorithms to co-adapt a simple controller architecture of agents trying to crouch forward as fast as possible. Similarly, earlier works by (Sims, 1994) used competition as a reward signal in a genetic algorithm to adapt the bodies of two robots fighting against each other in a virtual arena. Approaches for evolutionary robotics have been successfully applied to a number of different robotic platforms, primarily in simulation (Bongard, 2013), although recent works have identified that developing methods applicable to real world evolution remains an open challenge (Doncieux et al., 2015). Recent work has focused primarily on the fast changeability of robotic platforms as means to allow real world evolution of robots, such as extendable legs (Nygaard et al., 2021) or modularity (Hale et al., 2019; Alattas et al., 2019), although this constrains the range of possible robot designs considerably.

**Co-Adaptation with Reinforcement Learning:** Recent works have increasingly sought to improve data-efficiency and applicability of co-adaptation by using a reinforcement learning method as its main component. Seminal work by (Ha, 2019) introduced a policy gradient framework to jointly co-adapt the body and behaviour of agents in simulation with REINFORCE (Williams, 1992). (Schaff et al., 2019) extended this approach by proposing a deep reinforcement learning co-adaptation algorithm. Increased data-efficiency was achieved by (Luck et al., 2019) with the introduction of an off-policy deep reinforcement learning method using the Q-value function for design candidate evaluations. Another recent work (Gupta et al., 2021) employed deep reinforcement learning with mass-parallelization of agent populations in simulation, hence ignoring data-efficiency, using evolutionary techniques to investigate the Baldwin effect and Lamarckian evolution, for example.

**Imitation Learning:** Imitation learning has been a key technique in robot learning to enable agents to repeat behaviour demonstrated by humans (Fang et al., 2019; Asfour et al., 2008). Early techniques such as Behaviour Cloning (Pomerleau, 1988; Bain & Sammut, 1995) use a supervised learning strategy to extract motion policies replicating demonstrated behaviour. Generative Adversarial Imitation Learning (GAIL) (Ho & Ermon, 2016) measures the success of an imitator using an adversarial deep learning approach, employing a logistic loss to differentiate between the policies of the agent and the demonstrator. Other adversarial imitation learning algorithms have been devised in an attempt to perform well under changing state and action space representations, as well as different

transition functions. Adversarial Inverse Reinforcement Learning (AIRL) (Fu et al., 2018) produces disentangled rewards with respect to the environment dynamics. In contrast with the usage of the Jensen–Shannon divergence (Lin, 1991) in GAIL, State Alignment-based Imitation Learning (SAIL) (Liu et al., 2019) attempts to minimize the Wasserstein distance (Villani, 2009) between the state distributions induced by the demonstrator and the agent's policies. Closest to our work, (Rajani et al., 2023) proposed a first approach integrating morphological agnostic imitation learning into the co-adaptation process to adapt agent behaviour and design without an environmental reward and only given human expert demonstrations. Similarly, for our proposed method we include an imitation signal in the learning process. Crucially, however, CoSIL employs also the goal-oriented reward as primary objective for policy and design optimization, using imitation learning as secondary guidance to imitate the agent's previous behavior (i.e., self-imitation).

## 6 LIMITATIONS

While we can show that CoSIL increases the performance of co-adaptation with the help of a self-imitation reward, there are obvious limitations to this approach. We can argue that CoSIL increases data-efficiency and achieves higher performance with less morphologies, a key advantage given that the construction and manufacturing of robot prototypes in the real world is a costly and time-intensive endeavour. However, it is worth to point out that CoSIL adds a considerable computational overhead. In addition to multi-body reinforcement learning, CoSIL requires the costly training of discriminator networks in order to generate rewards via $r^{\text{IL}}$. In our experiments, we run CoSIL as long as possible on the available cluster infrastructure for a time duration of 72 hours. Standard co-adaptation with reinforcement learning (Coadapt) was capable of evaluating designs almost twice as fast than CoSIL; nonetheless, the converged performance of CoSIL was still higher. Hence, as we describe in our analysis about the limitations of CoSIL, one may not want to employ our proposed self-imitation learning approach on problems with low task complexity or low dimensionality in the morphology space as it is the case with the HalfCheetah task. Furthermore, our approach introduces another set of hyper-parameters, here the weights $\omega$ and $\omega_{\text{opt}}$, which may have to be fine-tuned for any given task. This could be alleviated in future work by introducing an automatic adaptation method.

## 7 CONCLUSION

We presented a new co-adaptation method named **Co**-Adaptation with **S**elf-**I**mitation **L**earning (CoSIL) which introduces the idea of using a self-imitation reward within a reward-driven co-adaptation framework using deep reinforcement learning for the purpose of jointly adapting the morphology and behaviour of embodied agents. To achieve this, we used State-Aligned Imitation Learning (SAIL) (Liu et al., 2019), introduced a method to select and match expert data from previously seen morphology-policy combinations, and employed separate Q-value functions for the objective and imitation rewards to increase data-efficiency when optimizing the morphology parameters. In experiments on morphology-adaptable agents in simulation, we showed that by imitating previously seen behaviour we can combat the distributional shift in dynamics, action and state spaces. Furthermore, we are able to demonstrate that self-imitation in combination with reward-driven co-adaptation can outperform both classical co-adaptation with rewards and pure imitation learning approaches. However, CoSIL requires a larger amount of computational effort due to additional deep neural network training, which makes it not preferable for simple co-adaptation problems. Nevertheless, with the methodology proposed in this paper we make a further step towards the useful integration of imitation learning techniques into co-adaptation techniques using deep reinforcement learning. Several interesting avenues for future work are opened up by our work, such as the use of quality-diversity approaches for selection of self-demonstrations, or further investigations of using a self-imitation reward during design optimization.

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

# A    IMPLEMENTATION DETAILS

In tables 2, 3 and 4, we provide the hyper-parameter values used throughout our experiments for CoSIL, SAC and SAIL, respectively. In Table 5, we specify the versions of the key Python packages we used to run these experiments. The code we developed to implement CoSIL and to perform our analysis is publicly available at [*censored URL for anonymity*].

Table 2: CoSIL hyper-parameters used in all experiments.

| Hyper-parameter | Value |
|---|---|
| Batch size | 256 |
| Replay buffer capacity | $2 \times 10^6$ |
| Number of episode demonstrations | $\{10,20,40\}$ |

Table 3: SAC hyper-parameters used in all experiments.

| Hyper-parameter | Value |
|---|---|
| $\gamma$ | 0.99 |
| $\tau$ | 0.005 |
| Learning rate | 0.0003 |
| $\alpha$ | 0.2 |
| Automatic entropy tuning | False |
| Hidden size of networks | 256 |
| Q-networks weight decay | $10^{-5}$ |

Table 4: SAIL hyper-parameters used in all experiments.

| Hyper-parameter | Value |
|---|---|
| Batch size | 64 |
| Normalization type | Z-score |
| Number of SAIL offline pre-training updates after a morphology change | $10^4$ |
| Learning rate | 0.0003 |
| Hidden size of the networks | 256 |
| Weight decay of the discriminator | $10^{-5}$ |
| Weight decay of the inverse dynamics model | $10^{-5}$ |

Table 5: Versioned Python software packages.

| Package | Version |
|---|---|
| gpy | 1.10.0 |
| gpyopt | 1.2.6 |
| gym | 0.26.2 |
| mujoco-py | 2.1.2.14 |
| numpy | 1.23.0 |
| pyswarms | 1.3.0 |
| python | 3.10.9 |
| torch | 1.13.1 |

# B    ENVIRONMENTS

In this section we give an overview of the environments used, inspired by previous environments proposed in Luck et al. (2019) and Rajani et al. (2023).

### B.1 HALFCHEETAH

We extend the standard HalfCheetah task to be morphological adaptable by allowing the change of lengths of the leg-segments. The original leg-lengths of HalfCheetah are $[.145, .15, .094, .133, .106, .07]$, where the first three numbers represent the lengths of the back leg, and the latter the lengths of the segments in the front leg. We allow the segment-lengths to be changeable in within the lower and upper bounds of $[x \cdot 0.2, x \cdot 2.0]$ for a length parameter $x$. The environmental reward function is given by

$$r^{\text{RL}} = \max \left( \frac{x_t - x_{t-1}}{\Delta t} - 0.1 \cdot |\mathbf{a}_t|_1^2, 0 \right), \tag{15}$$

where $x_t$ is the x-position of the torso and $\Delta t$ the simulation time-step. For HalfCheetah we train each morphology for 100 episodes and use $\omega = \omega_{\text{opt}} = 0.1$. As features we use the length-normalised position and velocity of the foot marker in respect to the base-length of the respective leg. In HalfCheetah we use a demonstration dataset of 10 trajectories/episodes.

### B.2 WALKER

For walker we adapt the morphological parameters (torso-length, leg-segment-top, leg-segment-bottom, foot-length) with the original parameters $[.6, .45, 0.5, .2]$. Similarly to HalfCheetah, these parameters are adaptable within the bounds of $[x \cdot 0.2, x \cdot 2.0]$ for a length parameter $x$. The environmental reward function is given by

$$r^{\text{RL}} = (\text{torso-height} > 0.5) \cdot \left( 1 + \frac{x_t - x_{t+1}}{\Delta t} \right) - 0.1 \cdot |\alpha|_2, \tag{16}$$

with $\alpha$ being the orientation of the Walker torso. For HalfCheetah we train each morphology for 200 episodes and use $\omega = \omega_{\text{opt}} = 0.2$. As features we use the length-normalised position and velocity of the foot marker in respect to the base-length of the respective leg. In Walker, we use a demonstration dataset of 20 episodes/trajectories.

### B.3 HUMANOID

In Humanoid we allow the symmetric adaptation of the parameters (thigh-length, shin-length, upper-arm-length, lower-arm-length), with the original parameters $[0.34, 0.3, 0.16, 0.16]$. These parameters are adaptable within the bounds of $[x \cdot 0.2, x \cdot 2.0]$ for a length parameter $x$. The reward function is given with

$$r^{\text{RL}} = 1.25(x_t - x_{t-1}) - 0.1|\mathbf{a}_t|_1^2 - \min(0.5 \times 10^{-6} \text{cfrc\_ext}_t^2, 10) + 5, \tag{17}$$

where $\text{cfrc\_ext}_t$ are the external forces acting on the body of the robot at timestep $t$. For Humanoid we train each morphology for either 300 or 1000 episodes, depending on the experiment, and use $\omega = \omega_{\text{opt}} = 0.2$ for CoSIL. As features we use the length-normalised position and velocity of the foot markers and hand markers in respect to the base-length of the respective leg or arm. In Walker, we use a demonstration dataset size 40 episodes/trajectories.

## C PERFORMANCE OF COSIL

As mentioned in the main paper, we show in Figure 4 the performance of each morphology sorted by its performance. This allows for a better comparison between CoSIL and baselines, as we found the morphology-optimisation process to be affected by the occasional miss-selection of the design optimisation process, something affecting both the baseline and CoSIL. We show the raw unsorted performance data of each morphology as encountered by the co-adaptation processes in Figure 7. It can be seen that while the mean performance is similar, standard deviations are noticeably increased due to the aforementioned effect. However, we find that CoSIL still outperforms the baseline. Figure 8 shows the progression of morphological parameters optimized by CoSIL in the two tasks Walker and Humanoid-300.

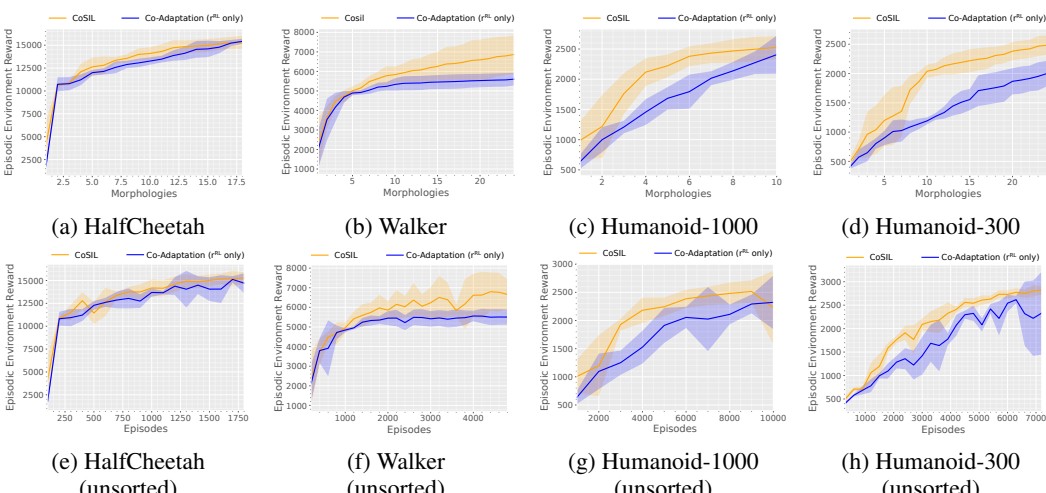

Figure 7: Comparison between our proposed approach CoSIL ($r^{IL}$ and $r^{RL}$) and Co-Adaptation (Luck et al., 2019) ($r^{RL}$ only) on the four tasks HalfCheetah, Walker, Humanoid-1000 and Humanoid-300 in MuJoCo. Plots show the performance of each morphology measured by averaging the 20% best episodes, and arranging the order of the morphologies by performance along the x-axis (see Appendix for plots without ordering). Experiments were repeated four times with distinct seeds. The top row (a-d) show the performance of each morphology evaluated from worst (left) to best (right). The bottom row (e-h) shows the performance of each morphology as encountered during the optimization process, and number of episodes evaluated. While each algorithm was trained for 1000 episodes on Humanoid-1000, in Humanoid-300 only 300 episodes were used. Comparing Fig. (c) and (d) shows that CoSIL increases the data-efficiency considerably when allowing for less episodes per morphology.

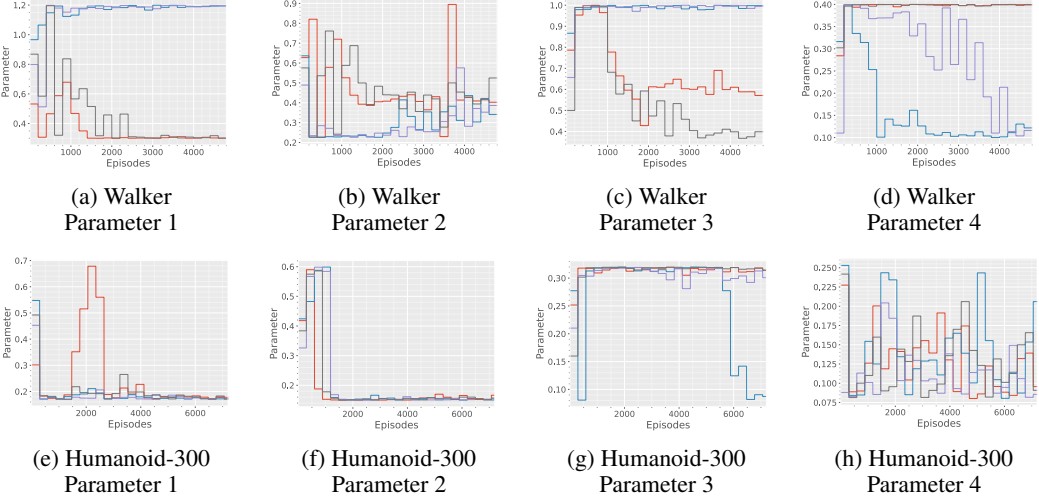

Figure 8: Progression of morphology parameters optimised by CoSIL for the two tasks Walker and Humanoid-300.

# D  CO-ADAPTATION OF UNITREE GO1 ROBOT

For further evaluation of the presented methodologies on a more challenging system we create a co-adaptable simulation of the Unitree Go1 quadruped as manufactured by Unitree Robotics. The model of the robot is based on URDF and CAD files provided by the Mujoco Menagerie. The robot has 12 degrees-of-freedom, with 3 force-controlled joints in each leg. We introduce five design variables in total: Four design variables $\xi_{1:4} \in [0.04, 0.4]$ influence the length of the bottom leg-segment of the robot, which is in contract with the ground. To further increase the difficulty of the task, we also allow the adaptation of the movement range of the top-most joint of the robot which is here an abduction joint, which can be changed with $\xi_5 \in [0.01, 0.8]$ radians for all four legs simultaneously. This introduces another change to the action and state spaces: Adapting this design variable allows for either a reduced or enhanced movement range of the abduction joint. Due to the increased complexity of the robot platform and difficulty, the following reward function was used to encourage stable, upright and forward locomotion

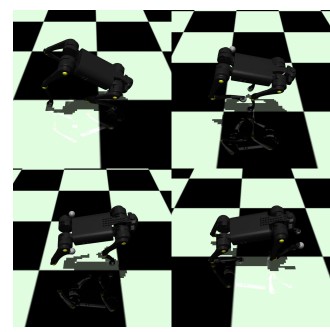

Figure 9: The simulated Unitree Go1 robot in the Mujoco Physics simulator performing forward locomotion when using the reward in Eq. (19).

$$\text{forward} = (0.5 + (h > h_{\text{init}} - 0.2) + (h > h_{\text{init}} - 0.1) \cdot 0.25 \tag{18}$$

$$+ (h > h_{\text{init}} \cdot 0.25))) \cdot \left( \frac{3.0 \cdot \Delta_x^+}{\Delta t} + 0.1 \right)$$

$$\text{upright} = -0.05 \cdot (|\alpha_y|^2 + |\alpha_x|) - 0.5 \cdot (|\alpha_y| > 1.0)$$

$$\text{control} = -0.001 \cdot \| \mathbf{a} \|_2$$

$$r^{\text{RL}} = \text{forward} + \text{upright} + \text{control}, \tag{19}$$

where $h$ is the curren height of the robot, $h_{\text{init}}$ the height of the robot when standing, $\Delta_x^+$ the positive displacement of the robot along the x-axis, $\Delta t$ the time between two steps, $\alpha_x$ the rotation of the robot along its x-axis, and $\alpha_y$ along its y-axis, both in radians. $\mathbf{a}$ is here the 12-dimensional action vector with $\mathbf{a} \in [-1, 1]^{12}$. For each selected morphology we evaluate 500 episodes, with each episode being 600 steps long at most. We used furthermore an early termination signal if the quadruped fell down, i.e. we terminated when $|\alpha_x| > 1.8$ or $h \leq h_{\text{init}} - 0.25$.

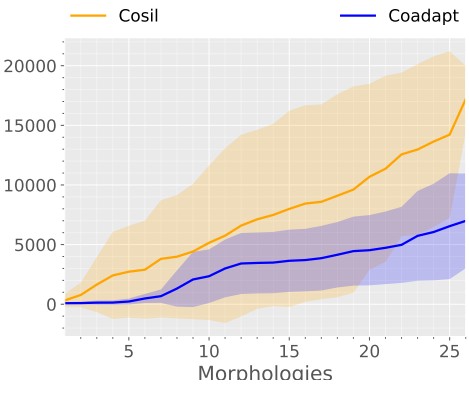

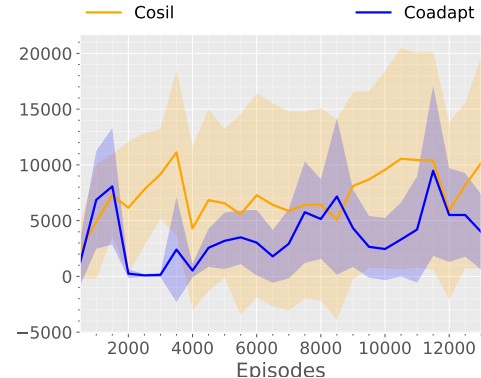

(a) Performances of discovered morphologies from worst (left) to best (right) for both methods.

(b) Performance of CoSIL and $r^{\text{RL}}$-only Co-adaptation throughout the co-adaptation process.

Figure 10: Performance of the proposed co-adaptation method utilizing self-imittaion learning (CoSIL, orange) versus co-adaptation without (Coadapt, blue, $r^{\text{RL}}$ only). It can be seen in both figures (a) and (b) that CoSIL is not only able to uncover more better-performing robot morphologies, but also outperforms objective-only-driven co-adaptation learning without being as affected by distributional shifts in action- and state-spaces as co-adaptation. Standard deviations and menas were computed over four experiments.

## D.1 RESULTS

Using the experimental setup of the Unitree robot in the Mujoco physics simulator we evaluate both the proposed co-adaptation method with self-imitation learning versus the standard reward-driven co-adaptation process. We performed for each methods four experiments with different seeds and allowed experiments to run for approximately 250 hours. The result confirm the previous experiments, that CoSIL shows better performance in more complex task and agent settings, such as humanoid and the Unitree Go1 robot. The results indicate that CoSIL is more resistant against the distributional shifts in action- and state-spaces when switching between morphologies (Fig. 10b). Furthermore, CoSIL is able to uncover better performing combinations of morphology and behaviour than reward-only-driven co-adaptation, highlighting the increased sample- and data-efficiency achievable with self-imitation learning in a co-adaptation setting.

