# OpenReview forum: "Sample-Efficient Co-Optimization of Agent Morphology and Policy with Self-Imitation Learning"
_ICLR.cc/2025/Conference — ICLR 2025 Conference Withdrawn Submission_

### Official Review · Reviewer_bH6J · 2024-11-01

**Soundness:** 2
**Presentation:** 2
**Contribution:** 2
**Rating:** 3
**Confidence:** 5

**Summary:**

In this paper, the authors proposed a new brain-body co-design method for embodied agents called CoSIL to co-optimize their morphologies and control policies simultaneously. SAIL combines reinforcement learning and State-Aligned Self Imitation Learning to improve the data-efficiency of the co-adaptation process as well as the behavioral recovery when adapting morphological parameters.

**Strengths:**

The research topic of the paper is interesting and the paper addresses the data efficiency problem on co-design.

**Weaknesses:**

1. This paper lacks a review and discussion of recent papers on the co-design of morphology and control for embodied agents.

2. The evaluation of the paper is not comprehensive. The experimental setup is limited to rigid-bodied robot.

3. Minor: line 245.

**Questions:**

1. Are there any other competitive brain-body co-design methods that can be compared? Such as Transform2Act, which utilizes GNN to parameterize the robot morphology.

2. While simulator such as MuJoCo is helpful to evaluate theories, how are these simulators, the proposed method, and the experimental results applicable to real robots?

3. What is the distributional shifts in the context of co-design problem?

---

### Official Review · Reviewer_Sg9v · 2024-11-02

**Soundness:** 2
**Presentation:** 2
**Contribution:** 2
**Rating:** 3
**Confidence:** 3

**Summary:**

The paper presents a co-adaptation methodology called Co-Adaptation with Self-Imitation Learning (CoSIL), which integrates reward-driven reinforcement learning with self-imitation learning to enhance the adaptation of both morphology and behavior in embodied agents. The authors try to address the challenge of catastrophic forgetting that can occur when agents with varying morphologies learn from each other, proposing a method that encourages agents to imitate their predecessors' behaviors to stabilize learning. By extending State-Aligned Imitation Learning (SAIL), the approach allows for adaptation when morphologies do not match.

**Strengths:**

* The idea of using imitation learning to encourage agents to imitate their predecessors' behaviors to stabilize learning is quite novel.
* The co-optimization of robot morphology and control policy is an important research question.

**Weaknesses:**

* The paper lacks a clear figure to illustrate the pipeline of the proposed method.
* More evaluation should be done in more challenging environments (like robots in 3D space, not just 2D). For example, Humanoid (3D version in Humanoid Walk or Humanoid Run tasks) and Quadruped.
* More co-adaptation baselines should be compared in the experiments, such as Transform2Act (https://arxiv.org/abs/2110.03659). This paper parametrizes the robots with GNN and also uses reinforcement learning to simultaneously explore the potential optimal morphology design and control policy.

**Questions:**

* What is the performance of the proposed method in more challenging 3D environments?
* What are the performance comparison results of the proposed method with other co-adaptation methods?

---

### Official Review · Reviewer_EP54 · 2024-11-02

**Soundness:** 3
**Presentation:** 3
**Contribution:** 2
**Rating:** 5
**Confidence:** 3

**Summary:**

This paper introduces Co-Adaptation with Self-Imitation Learning (CoSIL), a framework for training agents with different morphologies to learn by imitating each other. CoSIL combines environmental rewards with a self-imitation reward to guide learning, using State-Aligned Imitation Learning (SAIL) to match state distributions across morphologies. The approach leverages Soft Actor-Critic (SAC) for policy updates and Particle Swarm Optimization (PSO) for efficient morphology optimization. It's a nested optimization, PSO optimizes the morphology, and imitation learning is used to learn control policy.

**Strengths:**

The paper is well-written and easy to follow. The tests are performed in both simple robots and in complex systems. Moving the unitree experiments to the main paper will be better.

**Weaknesses:**

While the contributions are clear, I feel it is a little behind on the current developments in the co-design area. Several new co-design methods use RL to concurrently learn the morphology and behavior. Some of them use advanced networks to optimize both and have the capability to add or remove joints/limbs etc. while optimizing the parameters. While there is no mention of this add/remove capability, you could have at least compared it with any of those new methods. The method is deployed only in 3 environments which I think is insufficient to conclude. There are other environments such as Crawler, terrain crosser, swimmer, and glider. Comparing the performance in multiple environments helps us understand how the method can perform at varying numbers of morphology parameters and also helps us understand the growing computational needs. Some of the co-design papers that you can explore to find strong baselines are as follows,

1.	Agrim Gupta, Linxi Fan, Surya Ganguli, and Li Fei-Fei. Metamorph: Learning universal controllers with transformers. In International Conference on Learning Representations, 2021.
2.	Tingwu Wang, Yuhao Zhou, Sanja Fidler, and Jimmy Ba. Neural graph evolution: Towards efficient automatic robot design. In International Conference on Learning Representations, 2018.
3.	Ye Yuan, Yuda Song, Zhengyi Luo, Wen Sun, and Kris M. Kitani. Transform2act: Learning a transform-and-control policy for efficient agent design. In International Conference on Learning Representations, 2021.

**Questions:**

1. Using Q-functions as surrogate models for morphology optimization might introduce biases, like you mentioned in the paper, a small change in morphology can affect the policy largely. How accurate are these Q-functions in predicting the true performance of morphologies? Further analysis is required
2. How does CoSIL balance the exploration of new morphologies with the exploitation of already promising morphologies? Is there a mechanism to ensure sufficient exploration across the morphology space?

---

### Official Review · Reviewer_TfDG · 2024-11-02

**Soundness:** 2
**Presentation:** 3
**Contribution:** 2
**Rating:** 3
**Confidence:** 4

**Summary:**

This paper integrates concepts from State Alignment-based Imitation Learning [1] and Data-efficient Co-Adaptation of Morphology and Behavior with Deep Reinforcement Learning [2]. It introduces a co-adaptation method that combines Q-learning-based morphology optimization with State-Aligned Self-Imitation Learning, comparing its performance with the approaches in [1] and [2]. Unlike [2], which solely adapts body and behavior to maximize objectives like forward velocity, this method also encourages the agent to imitate the behaviors of its "ancestors," as proposed in [1].

References:
1. Liu, F., Ling, Z., Mu, T., & Su, H. (2020). State Alignment-based Imitation Learning. In Proceedings of the International Conference on Learning Representations (ICLR). https://iclr.cc/virtual_2020/poster_rylrdxHFDr.html

2. Luck, K. S., Amor, H. B., & Calandra, R. (2020). Data-efficient Co-Adaptation of Morphology and Behaviour with Deep Reinforcement Learning. In Conference on Robot Learning (CoRL). PMLR, pp. 854-869. http://proceedings.mlr.press/v100/luck20a/luck20a.pdf

**Strengths:**

Originality

The proposed method introduces self-imitation learning to evolutionary robotics in order to improve sample efficiency during co-optimization of morphology and behavior. Combining reinforcement learning with self-imitation is appealing, particularly in settings where real-world hardware constraints make high episode counts costly or infeasible.

Quality

The experiments seem to indicate that there is a small the acceleration afforded by CoSIL,  achieves superior performance with fewer episodes, achieving optimized policies and body configurations within 2,000–10,000 episodes.

Clarity

The paper is well-organized and clear, with each component of the proposed method described in detail.

Significance

The proposed method is interesting and inspiring for evolutionary robotics. Its emphasis on sample efficiency is a significant contribution to evolutionary robotics, addressing a core challenge for real-world applicability. By achieving efficient learning within a few thousand episodes, the proposed method has the potential to make adaptive co-optimization feasible for real-world robots.

**Weaknesses:**

Lack of experimental design, baselines, control experiments

The use of self-imitation learning for co-adaptation of an agent's morphology and control policy is the main contribution of the paper, but the experiments and evaluation lacks the control experiments and baselines necessary to understand the advantage of this approach. The only baseline presented is a model without self-imitation learning, making it difficult to assess whether self-imitation is indeed the most effective way to leverage prior training data for improving data efficiency. Additional comparisons with other reinforcement learning and imitation learning approaches (e.g. models using only behavioral cloning or adversarial imitation) would strengthen the evidence for CoSIL’s efficacy.

Hand-crafted feature space for imitation learning

The method’s reliance on hand-designed features (e.g. specific motion capture markers on the agent’s body) limits its practicality. Sect. 4.5 highlights how these features impact CoSIL’s performance on the HalfCheetah task, but it’s not clear how much this feature setup matters for other tasks. A more flexible approach, or experiments that show the effects of different feature spaces across tasks, could illuminate the versatility and applicability of the proposed method.

Small effect size, high variance, lack of evidence

Fig. 4a shows that the baseline method has the same performance as the proposed method, and the baseline also has a smaller standard deviation. The authors argue that this is because the task is too simple; the control experiments or additional baselines necessary to support this argument are missing. Also, only four independent trials were conducted (four different random seeds). Given the high variance, this seems insufficient. Statistical hypothesis testing is needed. Finally, Fig. 4 is arranged by the order of the morphologies by performance; Fig. 7 in the appendix better illustrates the training process and shows that the proposed method tends to get unstable at the end of training.

**Questions:**

1. Have you considered comparing CoSIL with other methods such as combination of RL-based reinforcement learning and behavior cloning / Generative Adversarial Imitation Learning? This could help demonstrate whether self-imitation is indeed the most effective way to leverage prior training data. Could including these alternative baselines offer additional insights into the conditions where self-imitation has the most impact?

2. Since CoSIL relies on manually designed features for imitation learning, do you have any plans to explore automated or more generalized feature selection methods? This could improve practicality and potentially expand the method’s adaptability across different tasks.

3. In Sect. 4.5, you show the impact of feature design on the HalfCheetah task. Could you discuss whether similar feature dependencies might arise in other environments, and how this might affect CoSIL's performance in more complex scenarios?

4. Could additional control experiments, benchmarks or a larger sample of random seeds provide further support for the claim that the task’s simplicity limits performance gains?

5. Could additional control experiments, benchmarks or a larger sample of random seeds show the robustness of the proposed methods over longer training periods and the statistic significance of the performance improvement compared to baseline methods?

6. Can this method be generalized to more complicated design space rather than just one that has different length of limbs?

---

### Note · Authors · 2024-11-29

I have read and agree with the venue's withdrawal policy on behalf of myself and my co-authors.